# Low Prevalence of HER2-Positive Breast Carcinomas among Screening Detected Breast Cancers

**DOI:** 10.3390/cancers12061578

**Published:** 2020-06-15

**Authors:** M. Ángeles López-García, Irene Carretero-Barrio, Belén Pérez-Míes, Miguel Chiva, Carolina Castilla, Begoña Vieites, José Palacios

**Affiliations:** 1Unidad de Anatomía Patológica, Hospital Universitario Virgen del Rocío, 41013 Sevilla, Spain; man.lopez.sspa@juntadeandalucia.es (M.Á.L.-G.); mb.vieites.sspa@juntadeandalucia.es (B.V.); 2Centro de Investigación Biomédica en Red de Cáncer (CIBERONC), Instituto de Salud Carlos III, 28029 Madrid, Spain; bperezm@salud.madrid.org; 3Pathology Department, Hospital Universitario Ramón y Cajal, 28034 Madrid, Spain; irene.carretero@salud.madrid.org; 4Universidad de Alcalá de Henares, 28801 Madrid, Spain; 5Breast Pathology Unit, Hospital Universitario Ramón y Cajal, 28034 Madrid, Spain; miguel.chiva@salud.madrid.org; 6IRyCIS, Instituto Ramón y Cajal de Investigación Sanitaria, Hospital Universitario Ramón y Cajal, 28034 Madrid, Spain; 7Radiology Department, Hospital Universitario Ramón y Cajal, 28034 Madrid, Spain; 8Nodo Biobanco Hospital Universitario Virgen del Rocío—Instituto de Biomedicina de Sevilla, Biobanco del SSPA, Unidad de Anatomía Patológica, Hospital Universitario Virgen del Rocío, 41013 Sevilla, Spain; carolina.castilla@juntadeandalucia.es

**Keywords:** breast cancer, screening, HER2, estrogen receptor, progesterone receptor, triple negative, luminal

## Abstract

Conflicting results have been reported regarding the prevalence of screen-detected human epidermal growth factor receptor 2 (HER2)-positive breast carcinomas and non-screen detected HER2-positive breast carcinomas. To address this issue, we evaluated the prevalence of HER2-positive breast carcinomas in two independent regional screening programs in Spain. The clinicopathologic and immunohistochemical characteristics of 479 (306 and 173) screen-detected breast carcinomas and 819 (479 and 340) non-screen-detected breast carcinomas diagnosed in women between 50 and 69-year-olds were compared. The prevalence of HER2-positive breast carcinomas was 8.8% and 6.4% in the two series of screen-detected tumors, compared with 16.4% and 13% in non-screen-detected carcinomas. These differences were statistically significant. This lower prevalence of HER2-positive in-screen-detected breast carcinomas was observed in both hormone receptor positive (luminal HER2) and hormone-receptor-negative (HER2 enriched) tumors. In addition, a lower prevalence of triple-negative and a higher prevalence of luminal-A breast carcinomas was observed in screen-detected tumors. Moreover, a literature review pointed out important differences in subrogate molecular types in screen-detected breast carcinomas among reported series, mainly due to study design, technical issues and racial differences.

## 1. Introduction

Since the introduction of screening programs for breast cancer (BC) detection, several studies have demonstrated that screen detected breast cancers (SDBC) are diagnosed in early stages, since they are smaller and have less lymph node metastases than those detected outside of screening [1,2,3,4,5]. In addition, SDBC have intrinsic good prognosis features, given that low-grade carcinomas and favorable types such as tubular carcinoma are more frequently diagnosed [3,6,7]. Partially due to these features, screening is associated with a relative mortality reduction of 20% [8].

During the last two decades, the introduction of the molecular classification of BC has prompted researchers to analyze the differences of the intrinsic molecular subtypes between SDBC and non-screen-detected breast cancers (NSDBC). Most studies have reported higher percentage of estrogen receptor (ER)-positive/progesterone receptor (PR)-positive tumors among SDBC leading to a higher prevalence of luminal A and a lower prevalence of triple negative (TN) breast carcinomas in this group of tumors [4,9,10,11]. However, there are conflicting results regarding the frequency of detected human epidermal growth factor receptor 2 (HER2)-positive BC among SDBC, since most series have not found differences when compared with NSDBC [7,10,11,12,13,14,15], while other studies have reported a lower prevalence of HER2-positive tumors within SDBC group [3,4,13,16]. These discrepancies can be attributed to different factors. Reported series are heterogeneous regarding to variables such as case and controls numbers, year and patient age at diagnosis, screening-program inclusion criteria or subrogate intrinsic molecular classification used.

Knowing the true frequency of HER2-positive BC among SDBC has an important clinical and epidemiological implication. To identify the correct proportion of HER2-positive tumors after benchmarking to standard references is an important measure to assurance quality in HER2 testing. As the detection mode seems to impact on HER2 status, the proportion of HER2-positive BC may differ in each institution depending on the relative number of tumors coming from screening programs. In addition, as currently available anti-HER2 drugs have improved HER2-positive tumors prognosis and their diagnosis is mostly outside the screening programs, there is probably an impact on the screening prognosis.

In this study, we establish the prevalence of the different subrogate intrinsic molecular subtypes in two independent hospital-based series of SDBC from two different regional screening programs in Spain, one from Andalusia (Programa de Detección Precoz del Cáncer de Mama (PDP)) and other from Madrid (Detección Precoz del Cáncer de Mama (DEPRECAM)). In both series, pathologic and immunohistochemical features were centrally reviewed and the control groups consisted of tumors from patients in the same age range, in order to avoid bias related to pathologic characteristics and/or age. Thereby, it is well established that the biologic characteristics of BC differ depending on the menopausal status of the patients. After an extensive literature review, our results in both series support the latest data indicating a low prevalence of HER2-positive among SDBC.

## 2. Results

### 2.1. Clinicopathologic Differences of NSDBC According to Age: Selection of the Control Group

First, in the hospital-based series from Andalusia, we analyzed possible differences among NSDBC according to age. We compared three age groups: ≤49 years, 50–69 years and ≥70 years. We observed statistically significant differences in all variables analyzed, except in PR expression (Table 1). Accordingly, we selected only NSDBC from women with 50–69 years for further comparisons with SDBC.

### 2.2. Clinicopathologic Differences between SDBC and NSDBC Groups

We observed statistically significant differences in all variables analyzed when compared SDBC with NSDBC (Table 2). Thus, near two thirds of SDBC were pT1 and N0, in contrast to only 50% in NSDBC. Regarding histological type, fewer infiltrating lobular carcinomas and poor-prognosis subtypes were found among SDBC. In addition, a two-fold increase in histological grade 1 was observed in SDBC with respect to NSDBC. In addition, 14.4% of SDBC showed lymphovascular invasion (LVI) in contrast with 33% of NSDBC.

### 2.3. Immunohistochemical and Subrogate Molecular Classification Differences between SDBC and NSDBC Groups

SDBC showed an increased proportion of ER-positive and PR-positive cases. The most important difference was observed regarding HER2 status. Whereas 16.4% of NSDBC were HER2-positive by immunohistochemistry (IHC) and/or fluorescence in situ hybridization (FISH), only 8.8% of SDBC were HER2-positive (*p* = 0.003) (Table 3).

According to biomarker results, we observed differences in the distribution of subrogate molecular types. The greater difference was observed in the frequency of HER2-positive/hormone receptor (HR)-negative BC, which dropped from 10% in NSDBC to 3.9% in SDBC. In addition, an increase in the frequency of luminal A and a decrease in the frequency of HER2-positive/HR-positive and TN BC was observed among SDBC. Regarding TN BC, the decrease was more prominent among those tumors that expressed basal markers. The proportion of luminal B BC was similar among SDBC and NSDBC (Table 4).

### 2.4. Prognosis

With a median follow-up of 120 months, we observed a total of 29 (9.4%) relapses and 16 deaths (5.22%) among SDBC (PDP) and 85 relapses (18%) and 75 deaths (15.9%) among NSDBC. However, deaths attributable to BC were 9 (2.9%) among SDBC and 48 (10.2%) among NSDBC. Kaplan-Meir plot is provided in Figure 1. Table 5 and Table 6 show the results of the univariate and multivariate analysis BC specific death. Analysis on time to relapse is provided in Appendix A. In the univariate analysis, SDBC had a better prognosis than NSDBC. However, in the multivariate analysis, screening did not retain statistical significance as a prognostic factor.

### 2.5. Literature Review

This review included 13 series (Table 7 and Table 8) in which immunohistochemical data and/or the subrogate molecular classification were evaluated.

The series reported by Joensuu et al. [14] and Lehtimäki et al. [15] included a similar cohort of patients, although the method of evaluation of HR differed between both studies. Series were heterogeneous with respect to different variables, as follows:

#### 2.5.1. Country

This review included eight series from Europe [3,4,10,12,13,14,15,18], three series from Asia [7,11,19], one series from United States of America [17] and one series from Oceania [20]. No series from Africa or South America were found.

#### 2.5.2. Year of Diagnosis

Year of diagnosis varied from 1985 to 2015, with most series analyzing cases during the first decade of this century [3,4,7,10,11,12,19,20].

#### 2.5.3. Age

Seven series [7,10,11,12,17,19,20] did not define an age-limit to include cases, whereas the remaining series limited the study to those included in the age range of the screening programs, which was variable among countries.

#### 2.5.4. Type of Cohort

Seven cohorts were composed of patients included in hospital-based databases, whereas the remaining series obtained the cases from population-based registries [3,13,14,15,18,20].

#### 2.5.5. Case Definition

SDBC was defined in 9 series as those tumors detected by mammography in patients attending population screening programs [3,4,7,12,13,14,15,18,20]. In two series the definition also included tumors detected by mammography outside screening programs [10,11]. Two series included patients with tumors detected by mammography, but none of them attending population screening programs [17,19].

#### 2.5.6. Centralized Review of Clinicopathologic Data

Only in one series, conventional clinicopathologic data were centrally reviewed [18]. In the remaining series, data were collected from medical records.

#### 2.5.7. Centralized Review of Immunohistochemistry Results

In seven series, IHC was specifically performed or evaluated for the study [3,10,13,14,15,17,18], whereas in the remaining six series immunohistochemical data were retrieved from medical records [4,7,11,12,19,20].

#### 2.5.8. ER Results

The threshold of positivity for ER was 1% in most series, but a 5% [17], 10% [4,7,18] or 20% [3] was also used in some studies. ER-positive cases ranged from 62.9% [11] to 86.7% [3] in NSDBC and from 69.4% [14] to 92.8% [3] in SDBC. Statistically significant differences in ER expression (a higher proportion of ER positive cases in SDBC) was observed in seven series [3,4,11,13,15,19,20], while three series did not find differences [7,10,14] and the data were missing in three series [12,17,18].

#### 2.5.9. PR Results

The threshold of positivity for PR was 1% in most series, but a 5% [17], 10% [4,7,18] or 20% [3] was also used in some studies. PR-positive cases ranged from 41.6% [18] to 77.5% [3] in NSDBC and from 46.7% [18] to 81% [3] in SDBC. Statistically significant differences in PR expression (a higher proportion of PR positive cases in SDBC) was observed in eight series [4,10,11,13,14,15,19,20], whereas two series did not find differences [3,7] and the data were missing in three series [12,17,18].

#### 2.5.10. HER2 Results

HER2-positive cases ranged from 12% [13] to 30.1% [7] in NSDBC and from 8% [13] to 25% [7] in SDBC. Statistically significant differences in HER2 expression and/or amplification (a lower proportion of HER2 positive cases in SDBC) was observed in only four series [3,4,19,20], while seven series did not find differences [7,10,11,12,13,14,15] and the data were missing in two series [17,18].

#### 2.5.11. Molecular Classification Definition

Different classification systems were used in different series. The most frequently used was: luminal A (ER-positive and/or PR-positive, HER2-negative) [10,11,13,20], luminal B (ER-positive and/or PR-positive, HER2-positive) [10,11,13,20], HER2-positive or HER2-enriched (ER-negative, PR-negative and HER2-positive) [3,10,11,13,19,20] and TN or basal-like (ER/PR-negative and HER2-negative) [3,4,10,11,13,17,19,20]. In some studies, luminal A and B were classified according to Ki-67 expression (>15% [17] or >20% [3,19] for luminal B tumors) and HR-positive HER2-positive cases were designed as luminal-HER2 [3,19].

#### 2.5.12. Molecular Classification Results

The results are summarized in Table 9. In control groups, the frequency of luminal (ER-positive and/or PR-positive, HER2-negative) tumors ranged from 54.3% to 72%; the frequency of luminal-HER2 (ER-positive and/or PR-positive, HER2-positive) tumors from 6% to 20.2%; the frequency of HER2-positive or HER2-enriched (ER-negative, PR-negative and HER2-positive) tumors from 3.5% to 13.9%; and the frequency of TN (ER/PR-negative and HER2-negative) tumors from 7.5% to 22.6%.

In SDBC, the frequency of luminal (ER-positive and/or PR-positive, HER2-negative) tumors ranged from 63.6% to 85%; the frequency of luminal-HER2 (ER-positive and/or PR-positive, HER2-positive) tumors from 5% to 16.7%; the frequency of HER2-positive or HER2-enriched (ER-negative, PR-negative and HER2-positive) tumors from 2% to 13%; and the frequency of TN (ER/PR-negative and HER2-negative) tumors from 1.8% to 18%.

## 3. Discussion

In this study, we confirmed that the frequency of subrogate molecular subtypes of BC differed between SDBC and NSDBC. Specially, we demonstrated that HER2-positive BC were underrepresented in SDBC, especially those tumors that were HR-negative. In addition—and according to most of the previously published series—we observed that SDBC showed statistically significant differences in most conventional clinicopathologic features analyzed when compared with NSDBC. Thus, SDBC presented at earlier stage, since they were smaller and had less axillary lymph node metastases. Regarding histological types, less infiltrating lobular carcinomas and poor-prognosis subtypes were found among SDBC. Moreover, the proportion of G1 carcinomas was higher in SDBC.

Biomarker expression results of previous series are conflicting, especially regarding the frequency of HER2-positive BC. Thus, whereas some studies did not find any differences [7,10,11,12,13,14,15], others observed [3,4,19,20], like the present study, a low prevalence of HER2-positive BC among SDBC. Disparities in patient selection, classification criteria and/or technical issues could explain the differences among series.

Regarding patient selection, one major strength of our study is that we used a control group composed by tumors affecting women in the same age range that those attending screening programs, since we demonstrated that clinicopathologic features differed among women with different ages. Thus, in both women younger than 49 years and older than 69 years, a lower proportion of stage I BC was observed due to larger tumors and more axillary involvement. In women younger than 49 years, a higher frequency of TN BC and a lower frequency of luminal A BC was observed. On the other hand, the group of women older than 69 years showed a lower proportion of HER2-positive BC.

Only three out of thirteen series reviewed used our same age range (50–69 years) [4,14,15] as a criterion for the control group and only in one of them the authors observed differences in HER2 expression regarding SDBC [4]. The reported lower prevalence of HER2-positive tumors within SDBC group in comparison with NSDBC group is probably related to a lead time bias due to early detection. Whereas the mean tumor size in HER2-positive SDBC in PDP series was 1.4 cm, a statistically significant higher mean size (2.65 cm) was observed in the non-screen detected HER2-positive BC group (*p* < 0.012).

The definition of case was also variable among the series. Thus, whereas we considered cases as only those detected during women’s participation in the population screening programs, other studies included tumors detected by mammography, without clinical symptoms, in patients from and outside screening programs [10,11]. Hence, a proportion of control tumors in our two cohorts could have been detected by mammography, without clinical symptoms. Whether or not these tumors have biologic characteristics more similar to SDBC or NSDBC remains to be established. In this sense, Iwamoto et al. [21] reported that the proportion of HER2-positive BC was 17% among “self-detected”, 15% among “screening-detected (asymptomatic)” and 15% among ‘‘screening-detected (symptomatic)’’.

Technical differences in the determination and evaluation of biomarkers were also present among series. Thus, different antibodies and thresholds criteria were used for the evaluation of ER, PR and HER2. Regarding ER and PR, some studied considered positive those tumors with at least 1% of positive cells, whereas others used a 5% [17], 10% [4] or 20% [3] threshold. For the evaluation of HER2, all but one [18] of the studies used the internationally accepted evaluation criteria. Probably these technical differences may partially explain the wide range of positivity of different biomarkers among series in both SDBC and NSDBC, as presented in Table 8. Regarding HER2, the reviewed series reported 12% to 30.1% of positivity in the control group and 8% to 25% of positivity in SDBC. Since we observed some differences in the incidence of HER2-positive BC between PDP and DEPRECAM series in our study, we carried out a concordance analysis in a small group of tumors with an overall kappa value for immunohistochemistry of 0.86 and 1 for in situ hybridization, corresponding to an almost perfect agreement. This concordance analysis suggested that the differences between both series were not due to technical issues (Appendix A). Our results in the NSDBC group are in accordance with national data in Spain, where a central data base [22] including 135,173 cases, reported a 16.4% of HER2 positivity. In addition, in a database from UK and Ireland, the frequency of HER2 positivity among 1.537 tumors was 9% in SDBC and 13.3% in symptomatic patients [23].

Regarding TN BC, most previous series demonstrated a reduction of this molecular type among SDBC [3,4,10,11,13,17,19,20]. However, there were marked differences in the relative frequency of this molecular type in SDBC, ranging from 1.8% to 18% in the different series. In addition to the previously mentioned factors, race can partially explain these differences, since a higher percentage was observed in that series including a large proportion of Black-American women, a population group in which TN BC is more frequent [24]. Only two previous series analyzed differences in the expression of basal markers among TN SDBC [10,13]. Our results, as the study of Crispo et al. [10], suggested a lower percentage of basal-like TN BC in this group. In contrast, Dawson et al. [13] did not observed this difference. Further studies are needed in order to establish whether or not different subtypes of TN BC are differentially represented between SDBC and NSDBC.

Most previous series have classified luminal tumors according to HER2 expression, and only three series have also classified luminal tumors according to Ki67 expression [3,17,19]. In our study, and the one from Brewster et al. [17], we defined luminal A carcinomas as those ER and/or PR positive BC that had Ki67 expression in ≤15% of neoplastic cells. Falck et al. [3] defined luminal tumors according to both Ki67 (20% threshold) and PR (20% threshold) expression. Despite differences in definition criteria, the three series observed a similar frequency of luminal B BC in SDBC and NSDBC, but a significant higher percentage of luminal A BC in the SDBC group. These data, together with the lower prevalence of HER2-positive and TN BC indicated that screening increased the proportion of low proliferative BC. Accordingly, we observed differences in Ki67 proliferative index among subrogate molecular groups (Appendix A).

Some studies have reported that screening is a favorable prognostic factor in BC. Other authors, however, have suggested that prognosis in this group of patients is related with associated clinicopathologic good prognostic factors. In this sense, in our series, patients with SDBC had better prognosis that patients with NSDBC in the univariate analysis. However, screening status did not show prognostic significance in the multivariate analysis (Table 6). On the other hand, tumor size, stage and. Higher tumor size and more advanced stage were associated with worse outcomes. These variables were also important when evaluating HER2-positive BC only. In this subgroup of tumors, the method of detection was not associated with the prognosis, whereas tumor size and stage were (Appendix A).

In spite of HER2 status not being associated with prognosis, the molecular subtype was. Luminal A molecular subtype had a better disease-free survival and BC-specific survival, whereas basal BC had a worse disease-free survival and BC-specific survival. These results highlight the importance of using molecular classification on BC, as it can provide key prognosis information.

Our review pointed out important differences in subrogate molecular types in SDBC among reported series, partially attributable to differences in study designs, technical issues and racial differences. To know the true incidence of intrinsic molecular subtypes of SDBC is of great importance to increase epidemiological knowledge and to stablish benchmarks for quality controls, which could ultimately impact on patients’ treatment.

## 4. Materials and Methods

### 4.1. Patients and Tumors

#### 4.1.1. Cohort 1 (Programa de Detección Precoz (PDP) del Cáncer de Mama, Andalusia)

This cohort included all invasive BC diagnosed in the pathology Department of the Hospital Universitario Virgen del Rocío, Seville, Spain, between 2006–2009 in which at least one adequate paraffin block was available for tissue microarray (TMA) building. The series included 1089 tumors: 306 were detected in the Programa de Detección Precoz (PDP) del Cáncer de Mama de Andalucía and the remaining 783 were diagnosed outside the screening program. All histological sections were reviewed by two Pathologists (MAL-G and JP) for histological typing and grading, according to WHO recommendations (2013) and for the presence of LVI. The remaining pathologic features (tumor size and lymph node metastases) were obtained from the pathology records. Clinical records were reviewed to obtain the age of the patient, tumor location, type of treatment and outcome.

#### 4.1.2. Cohort 2 (Detección Precoz del Cáncer de Mama (DEPRECAM), Madrid)

This cohort included all invasive BC diagnosed in the pathology Department of Hospital Universitario Ramón y Cajal, Madrid, Spain, between 2013–2017. The series included 173 cases detected by the screening program “Detección Precoz del Cáncer de Mama” (Madrid) (DEPRECAM) and 340 cased detected outside the screening program. All histological sections were reviewed by two Pathologists (BP-M and JP) for histological typing and grading, according to WHO recommendations (2013).

### 4.2. Immunohistochemistry and Tumor Classification

IHC in Cohort 1 was analyzed on TMA. TMA construction was performed as previously reported [25] including two cores for each tumor. Analysis of TMA IHC and fluorescent in situ hybridization was performed by two pathologists (MAL-G and JP) and the final results were obtained by consensus.

HER2 in Cohort 2 was re-evaluated on core biopsies by two pathologists (BP-M and JP) and the final results were obtained by consensus. On HER2-positive cases, ER and PR were also evaluated. Details of the antibody clones, probes, suppliers, dilutions and scoring criteria following international guidelines [26] used are provided in Appendix A.

Since we used different reagents in PDP and DEPRECAM series to analyze HER2 status, we performed a concordance analysis between the two immunohistochemical methods using 2 TMA sections, including a total of 58 cases, displaying different levels of HER2 expression that were double-blinded evaluated.

For molecular classification, BC were grouped according to immunohistochemical criteria. As a subrogate definition of the intrinsic subtypes of BC the following criteria were applied [27]: tumors showing ER or PR expression and no HER2 expression were regarded as luminal A or B depending on their Ki67 index (≤15 or >15, respectively); tumors with ER or PR positivity and HER2 overexpression and/or amplification were classified as ER-positive/PR-positive/HER2-positive (luminal HER2); tumors showing overexpression and/or amplification of HER2 and absence of expression of ER and PR were regarded as HER2-positive and TN tumors with CK5/6, CK14, CK17 and/or EGFR expression were ascribed to the basal phenotype, whereas all other TN breast carcinomas were termed TN not otherwise specified (NOS).

### 4.3. Statistical Analysis

The χ^2^ test was used to evaluate differences among clinicopathologic characteristics (summarized with percentages). Differences in median size (in cm) among the different groups of study were analyzed with the Kruskal–Wallis and Mann–Whitney U tests. Breast cancer-specific survival was defined as the time from surgery to the time of death from BC, with deaths from other causes being censored; whereas in the time to relapse analysis, the endpoint was BC recurrence, either local or distant. Survival curves were estimated using the Kaplan–Meier method the differences in survival were evaluated using the log-rank test. Cox’s proportional hazards modeling of parameters potentially related to survival was conducted to calculate hazard ratios in both univariate and multivariate analyses.

Kappa values were used to assess HER2 status concordance between the results obtained by the two immunohistochemistry methods applied in this study.

All of these statistical analyses were performed using SPSS version 20 (SPSS, Inc.) and JMP 10 statistical software (SAS Institute, Inc., Cary, NC, USA). *p* < 0.05 was considered statistically significant.

The present study was performed in accordance with the standard ethical procedures dictated by Spanish law (Ley de Investigación Orgánica Biomédica, 14 July 2007). The procedure was approved by the local ethics committee (Code 0292-N-15, approval date 10 April 2015, by Comité de Ética de la Investigación-HHUU Virgen Macarena-Virgen del Rocío). Written informed consent was obtained from all the patients and all the clinical analyses were conducted in accordance with the principles of the Helsinki Declaration.

### 4.4. Literature Review

A literature review was conducted using the MEDLINE and PubMed. The following keywords were used to perform flexible searches within these databases: “screening” AND “breast cancer”, “pathology”, “immunohistochemistry”, “HER2”, “estrogen receptor”, “progesterone receptor”, “Ki67”, “molecular classification”. Only papers published in English were included.

## 5. Conclusions

In our series, HER2-positive BC are underrepresented in SDBC compared to NSDBC, particularly HR-negative tumors. In addition, within SDBC there is a higher prevalence of luminal A BC and a lower prevalence of TN BC. The literature review pointed out important differences in biomarker expression results as well as in subrogate molecular types, partially attributable to differences in study designs, technical issues and racial differences.

## Figures and Tables

**Figure 1 cancers-12-01578-f001:**
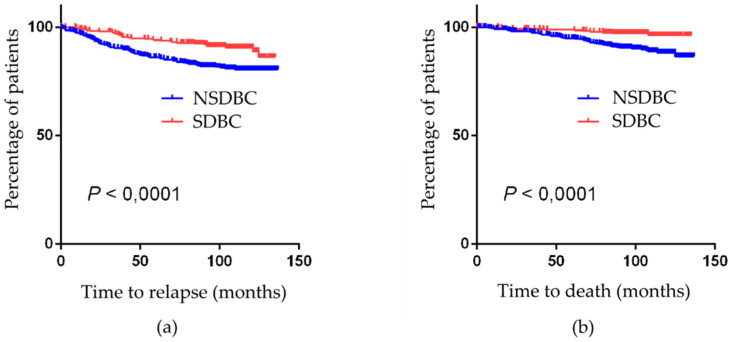
Kaplan–Meier plots according to mode of breast cancer detection: (**a**) Time to relapse (local and/or distant recurrence); (**b**) time-to-breast-cancer-specific death. Vertical spikes represent censored patients. *p*-values from Mantel–Cox log-rank test. NSDBC—non-screen-detected breast cancers, SDBC—screen-detected breast cancers.

**Table 1 cancers-12-01578-t001:** Pathologic features of breast carcinomas according to age.

Size (cm) (Median (p25–p75))	No of Patients	≤49 Years, *n* (%)	50–69 Years, *n* (%)	≥70 Years, *n* (%)	*p*
		2.2 (1.6–3.2)	2.1 (1.5–3)	2.8 (1.9–3.8)	<0.0001 ^b^
Size (pT)					
1	333	62 (39.5)	223 (48.2)	48 (30.8)	<0.0001 **^a^**
2	355	73 (46.5)	195 (42.1)	87 (55.8)	
3	48	11 (7)	29 (6.3)	8 (5.1)	
4	35	8 (5.1)	14 (3)	13 (8.3)	
Lymph node involvement (*N*)					
0	305	68 (43.3)	246 (53.1)	59 (42.1)	<0.0001 **^a^**
1	215	48 (30.6)	119 (25.7)	48 (34.3)	
2	93	21 (13.4)	55 (11.9)	17 (12.1)	
3	79	20 (12.7)	43 (9.3)	16 (11.4)	
Stage					
I	410	32 (20.6)	348 (45.2)	30 (19.4)	<0.0001 **^a^**
II	361	73 (47.1)	282 (36.7)	79 (50.9)	
III	163	42 (27.2)	123 (16)	40 (25.8)	
IV	30	8 (5.1)	16 (2.1)	6 (3.9)	
Histologic grade					
1	83	13 (8.3)	59 (12.6)	11 (7.1)	<0.0001 **^a^**
2	436	77 (49)	264 (56.2)	95 (60.8)	
3	264	67 (42.7)	147 (31.3)	50 (32.1)	
LVI					
Yes	269	63 (40.1)	155 (33)	51 (33.1)	0.001 **^a^**
No	510	93 (59.9)	314 (67)	103 (66.9)	
ER					
Positive	596	106 (67.5)	361 (76.8)	129 (82.7)	0.001 **^a^**
Negative	187	51 (32.5)	109 (23.2)	27 (17.3)	
PR					
Positive	508	99 (63.1)	299 (63.6)	110 (70.5)	0.3 **^a^**
Negative	275	58 (36.9)	171 (36.4)	46 (29.5)	
HER2					
Positive	120	30 (19.1)	77 (16.4)	13 (8.3)	0.02 **^a^**
Negative	663	127 (80.9)	393 (83.6)	143 (91.7)	
Phenotypes					
luminal A	310	45 (28.7)	201 (42.8)	64 (41)	<0.0001 **^a^**
luminal B	233	48 (30.6)	130 (27.7)	55 (35.3)	
luminal HER2	53	13 (8.3)	30 (6.4)	10 (6.4)	
HER2	67	17 (10.8)	47 (10)	3 (1.9)	
TN NOS	56	17 (10.8)	28 (6.9)	11 (7.1)	
basal	64	17 (10.8)	34 (7.2)	13 (8.3)	

^a^*p*-value from chi-squared score; ^b^*p*-value from Kruskal–Wallis test; LVI—lymphovascular invasion; ER—estrogen receptor, PR—progesterone receptor, HER2—human epidermal growth factor receptor 2, luminal A—ER/PR-positive, Ki67 ≤ 15%, HER2-negative; luminal B—ER/PR-positive, Ki67 > 15%, HER2-negative; luminal HER2—ER/PR-positive, HER2-positive; HER2—ER/PR-negative, HER2 positive; TN NOS—ER/PR-negative, HER2-negative, CK5/6/CK17/CK14/EGFR-negative; basal—ER/PR-negative, HER2-negative, CK5/6/CK17/CK14 and/or EGFR-positive.

**Table 2 cancers-12-01578-t002:** Histopathologic differences between screen detected breast cancers (SDBC) (Programa de Detección Precoz del Cáncer de Mama (PDP)) and non-screen-detected breast cancers (NSDBC).

Size (cm) (Median (p25–p75))	No of Patients	NSDBC, *n* (%)	SDBC, *n* (%)	*p*
	766	2.1 (1.5–3)	1.5 (1–2.05)	<0.0001 ^b^
Size (pT)				<0.0001 ^a^
1	452	223 (48.2)	229 (74.8)	
2	268	195 (42.1)	73 (23.9)	
3	32	29 (6.3)	3 (1)	
4	14	14 (3)	0 (0)	
Lymph node involvement (pN)				<0.0001 ^a^
0	472	246 (53.1)	226 (73.9)	
1	173	119 (25.7)	54 (17.6)	
2	74	55 (11.9)	19 (6.2)	
3	50	43 (9.3)	7 (2.3)	
Stage				<0.0001 ^a^
IA	341	151 (31.1)	190 (61.8)	
IB	13	10 (2.1)	3 (1)	
IIA	198	133 (28.4)	65 (21.2)	
IIB	84	64 (13.6)	20 (6.4)	
IIIA	67	48 (10.2)	19 (6.2)	
IIIB	10	9 (1.9)	1 (0.3)	
IIIC	46	39 (8.3)	7 (2.39	
IV	16	15 (3.2)	1 (0.3)	
Grade				<0.0001 ^a^
1	132	59 (12.6)	73 (23.9)	
2	432	264 (56.2)	168 (54.9)	
3	212	147 (31.3)	64 (21.2)	
LVI				<0.0001 ^a^
Yes	199	155 (33)	44 (14.4)	
No	576	314 (67)	262 (85.6)	

^a^*p*-value from chi-squared score; ^b^*p*-value from Mann–Whitney U test; LVI—lymphovascular invasion.

**Table 3 cancers-12-01578-t003:** Immunohistochemical differences between NSDBC and SDBC (PDP).

	No of Patients	NSDBC, *n* (%)	SDBC, *n* (%)	*p* ^a^
ER				0.001
Positive	626	361 (76.8)	265 (86.6)	
Negative	150	109 (23.2)	41 (13.4)	
PR				0.002
Positive	526	299 (63.6)	227 (74.2)	
Negative	250	171 (36.4)	79 (25.8)	
HER2				0.003
Positive	104	77 (16.4)	27 (8.8)	
Negative	672	393 (83.6)	279 (91.2)	

^a^*p*-value from chi-squared.

**Table 4 cancers-12-01578-t004:** Subrogate molecular classification of NSDBC and SDBC (PDP).

	No of Patients	NSDBC, *n* (%)	SDBC, *n* (%)	*p* ^a^
luminal A	361	201 (42.8)	160 (52.3)	0.005
luminal B	220	130 (27.7)	90 (29.4)	
luminal HER2	45	30 (6.4)	15 (4.9)	
HER2	59	47 (10)	12 (3.9)	
TN NOS	46	28 (6.9)	18 (5.9)	
basal	45	34 (7.2)	11 (3.6)	

^a^*p*-value from chi-squared; luminal A—ER/PR-positive, Ki67 ≤ 15%, HER2-negative; luminal B—ER/PR-positive, Ki67 > 15%, HER2-negative; luminal HER2—ER/PR-positive, HER2-positive; HER2—ER/PR-negative, HER2 positive; TN NOS—ER/PR-negative, HER2-negative, CK5/6/CK17/CK14/EGFR-negative; basal—ER/PR-negative, HER2-negative, CK5/6/CK17/CK14 and/or EGFR-positive.

**Table 5 cancers-12-01578-t005:** Tumor-associated death in the age range of 50–69-years old.

	No of Patients	Event, *n* (%)	Mean Time to Death (Months), (CI 95%)	*p* ^a^
Size (pT)				<0.0001
T1	452	10 (2.2)	134.2 (133.1–135.4)	
T2	268	29 (10.8)	126.1 (122.3–129.1)	
T3-T4	46	16 (34.7)	101.1 (86.9–115.3)	
Node involvement (pN)				<0.0001
N0	472	13 (2.7)	133.6 (132.4–134.8)	
N1	173	8 (4.6)	130.5 (101.6–133.4)	
N2,3	124	35 (28.2)	111.4 (104.1–118.7)	
Stage				<0.0001
I–II	636	15 (2.3)	133.8 (132.7–134.8)	
III–IV	139	41 (29.4)	109.8 (102.8–116.8)	
Grade				<0.0001
1–2	564	27 (4.7)	132.9 (131.5–134.4)	
3	212	30 (14.1)	119.1 (114.7–123.5)	
LVI				<0.0001
No	576	24 (4.1)	132.4 (133.8–131.7)	
Yes	199	33 (16.5)	121.7 (116.8–126.5)	
Relapse				<0.0001
No	662	3 (0.4)	136.4 (135.9–136.8)	
Yes	114	54 (47.3)	92.3 (84.1–100.5)	
SDBC (PDP)	306	9 (2.9)	132.6 (131–134.3)	<0.0001
NSDBC	470	48 (10.2)	127.7 (125.2–130.2)	
ER				0.006
Positive	626	39 (6.2)	131.9 (130.4–133.4)	
Negative	150	18 (12)	122.6 (117.02–128.7)	
PR				0.001
Positive	527	28 (5.3)	132.5 (131–134.1)	
Negative	249	29 (11.6)	124.4 (120.5–128.2)	
HER2				0.32
Positive	104	10 (9.6)	126.2 (120.7–131.7)	
Negative	672	47 (7)	130.8 (129.1–132.5)	
luminal A	361	9 (2.4)	134.2 (133–135.3)	<0.0001
luminal B	220	26 (11.8)	127.5 (124.1–130.9)	
luminal HER2	45	4 (8.8)	123.5 (116.5–130.4)	
HER2	59	6 (10.1)	125.1 (117.2–133)	
TN NOS	46	5 (10.8)	121.8 (112.6–131.1)	
basal	45	7 (15.5)	113.7 (101.7–125.6)	

^a^*p*-values from Mantel–Cox log-rank test; CI—confidence interval; luminal A—ER/PR-positive, Ki67 ≤ 15%, HER2-negative; luminal B—ER/PR-positive, Ki67 > 15%, HER2-negative; luminal HER2—ER/PR-positive, HER2-positive; HER2—ER/PR-negative, HER2 positive; TN NOS—ER/PR-negative, HER2-negative, CK5/6/CK17/CK14/EGFR-negative; basal—ER/PR-negative, HER2-negative, CK5/6/CK17/CK14 and/or EGFR-positive.

**Table 6 cancers-12-01578-t006:** Cox model for histopathologic and immunophenotypic features for disease-free survival and cancer-specific-free survival.

	Disease-Free Survival			Cancer-Specific Free Survival		
	Hazard Ratio	CI 95%	*p* ^a^	Hazard Ratio	CI 95%	*p* ^a^
Size (pT)						
T1	1			1		
T2	1.18	0.75–1.93	0.42	2.11	0.96–4.63	0.06
T3–T4	2.28	1.13–4.40	0.02	3.5	1.35–9.26	0.01
Node involvement (pN)						
N0	1			1		
N1	2.04	1.18–3.53	0.008	1.27	0.61–2.49	0.62
N2–3	1.74	1.12–3.48	0.03	1.05	0.52–2.99	0.89
Stage						
I-II	1			1		
III-VI	4.457	2.74–9.35	0.002	8.54	2.41–12.4	<0.0001
SDBC (PDP)	1			1		
NSDBC	1.12	0.70–1.78	0.62	1.69	0.79–3.64	0.17
Molecular subtypes						
luminal A	1			1		
Other	1.892	1.14–3.21	0.003	3.83	1.72–8.37	0.001
Basal	3.252	1.23–8.54	<0.0001	6.92	2.29–15.7	0.001

^a^ Cox’s proportional hazards modeling; CI—confidence interval; luminal A—ER/PR-positive, Ki67 ≤ 15%, HER2-negative; basal—ER/PR-negative, HER2-negative, CK5/6/CK17/CK14 and/or EGFR-positive.

**Table 7 cancers-12-01578-t007:** Features of the series.

Series	Country	Years of Diagnosis	Age	Type of Cohort	SDBC from Population Screening	Central Pathologic Review	Central IHC Review	SDBC Patients	NSDBC Patients
Joensuu 2004 [14]	Finland	1991–1992	50–69	Population–based	Yes	No	Yes	379	538
Palka 2008 [12]	Hungary	2004–2007	All	Hospital-based	Yes	No	No	255	262
Chuwa 2009 [7]	Singapore	2002–2003	All	Hospital-based	Yes	No	No	103	664
Dawson 2009 [13]	United Kingdom		50–70	Population-based	Yes	No	Yes	610	796
Brewster 2011 [17]	United States of America	1985–2000	≥40	Hospital-based	No	No	Yes	603	247
Lehtimäki 2011 [15]	Finland	1991–1992	50–69	Population-based	Yes	No	Yes	347	502
Olsson 2011 [18]	Sweden	1991–1996	47–75	Population-based	Yes	Yes	Yes	262	204
Kim 2012 [11]	South Korea	2002–2008	All	Hospital-based	Yes + mammography outside screening	No	No	1025	2116
Crispo 2013 [10]	Italy	2004–2006	All	Hospital-based	Yes + mammography outside screening	No	Yes	114	334
Domingo 2013 [4]	Spain	1995–2008	50–69	Hospital-based	Yes	No	No	97	97
Falck 2016 [3]	Sweden	1999–2003	45–74	Population-based	Yes	No	Yes	205	229
Kobayashi 2017 [19]	Japan	2003–2014	All	Hospital-based	No	No	No	274	858
Farshid 2018 [20]	Australia & New Zealand	2005–2015	All	Population-based	Yes	No	No	32,493	66,907
PDP	Spain	2006–2009	50–69	Hospital-based	Yes	Yes	Yes	306	470
DEPRECAM	Spain	2013–2017	50–69	Hospital-based	Yes	Yes	Yes	173	340

IHC: immunohistochemistry, PDP: Programa de Detección Precoz del Cáncer de Mama (Andalusia), DEPRECAM: Detección Precoz del Cáncer de Mama (Madrid).

**Table 8 cancers-12-01578-t008:** Proportion of immunohistochemistry positive cases in the series.

Series	ER		PR		HER2	
	SDBC	NSDBC	SDBC	NSDBC	SDBC	NSDBC
Joensuu 2004 [14]	69.4%	68%	62.5% ^a^	51%	15.5%	19.00%
Palka 2008 [12]					13%	16%
Chuwa 2009 [7]	71.2%	66.7%	60.6%	51.1%	25%	30.1%
Dawson 2009 [13]	86% ^a^	74%	74% ^a^	65%	8%	12%
Brewster 2011 [17]					11.7% ^b^	18.9%
Lehtimäki 2011 [15]	85% ^a^	67%	63% ^a^	50%	18%	19%
Olsson 2011 [18]	88.5% ^b^	85%	46.7% ^b^	41.6%	12.1% ^b^	16%
Kim 2012 [11]	72.1% ^a^	62.9%	65.9% ^a^	57%	21.9%	24.4%
Crispo 2013 [10]	72.2%	70.7%	78.1% ^a^	68%	14.8%	15.7%
Domingo 2013 [4]	87% ^a^	75.6%	68.5% ^a^	55%	12.6% ^a^	28.2%
Falck 2016 [3]	92.8% ^a^	86.7%	81%	77.5%	14% ^a^	24%
Kobayashi 2017 [19]	86.2% ^a^	74.7%	77% ^a^	72.5%	10.6% ^a^	15.9%
Farshid 2018 [20]	89.3% ^a^	80.3%	78.8% ^a^	69.8%	11.0% ^a^	15.6%
PDP	86.6% ^a^	76.8%	74.20% ^a^	63.60%	8.8% ^a^	16.4%
DEPRECAM					6.4% ^a^	13.0%

^a^ Statistically significant difference; ^b^ Statistical significance not provided; PDP—Programa de Detección Precoz del Cáncer de Mama (Andalusia), DEPRECAM—Detección Precoz del Cáncer de Mama (Madrid).

**Table 9 cancers-12-01578-t009:** Molecular classification results.

Series	SDBC, *n* (%)	NSDBC, *n* (%)	*p*
Dawson 2009 [13]			
luminal	322 (85)	366 (72)	<0.0001
luminal HER2	21 (5)	32 (6)	0.62
HER2	12 (3)	24 (5)	0.216
TN	26 (7)	84 (17)	
NOS	7 (2)	28 (6)	0.005
basal-like	19 (5)	56 (11)	0.001
Brewster 2011 [17]			<0.001
luminal	162 (65.6)	318 (52.7)	
luminal A	114 (46.2)	170 (28.2)	
luminal B	48 (19.4)	148 (24.5)	
TN	28 (11.3)	126 (20.9)	
Kim 2012 [11]			<0.0001
luminal	619 (63.6)	1.093 (54.3)	
luminal HER2	96 (9.9)	212 (10.5)	
HER2	117 (12.0)	279 (13.9)	
TN	142 (14.6)	429 (21.3)	
Crispo 2013 [10]			0.04
luminal	74 (68.5)	193 (59.0)	0.09
luminal HER2	18 (16.7)	59 (18.2)	0.71
HER2	14 (13.0)	38 (11.0)	0.79
TN	2 (1.8)	34 (10.5)	
NOS	2 (1.8)	18 (5.6)	0.01
basal-like	0	16 (4.9)	
Domingo 2013 [4]			
HER2	7 (8.1)	10 (13.3)	
TN	3 (3.5)	8 (10.7)	0.002
Falck 2016 [3]			0.011
luminal	160 (81.2)	119 (68.7)	
luminal A	92 (46.7)	62 (35.8)	
luminal B	68 (34.5)	57 (32.9)	
luminal HER2	23 (11.7)	35 (20.2)	
HER2	4 (2.0)	6 (3.5)	
TN	10 (5.1)	13 (7.5)	
Kobayashi 2017 [19]			<0.001
luminal	223 (81.4)	586 (68.3)	
luminal A	168 (61.3)	379 (44.2)	
luminal B	55 (20.1)	207 (24.1)	
luminal HER2	18 (6.6)	67 (7.8)	
HER2	11 (4.0)	69 (8.0)	
TN	22 (8.0)	136 (15.9)	
Farshid 2018 [20]			
luminal	(82.3)	(71.4)	<0.0001
luminal HER2	(7.6)	(9.8)	<0.0001
HER2	(3.5)	(5.8)	<0.0001
TN	(6.7)	(13.0)	<0.0001
PDP			
luminal	250 (81.7)	331 (70.5)	
luminal A	160 (52.3)	201 (42.8)	
luminal B	90 (29.4)	130 (27.7)	
luminal HER2	15 (4.9)	30 (6.4)	
HER2	12 (3.9)	47 (10)	
TN	29 (9.5)	62 (14.1)	
NOS	18 (5.9)	28 (6.9)	
basal-like	11 (3.6)	34 (7.2)	
DEPRECAM			
luminal HER2	6 (3.47)	43 (12.65)	
HER2	5 (2.89)	25 (7.35)	

PDP—Programa de Detección Precoz del Cáncer de Mama (Andalusia), DEPRECAM—Detección Precoz del Cáncer de Mama (Madrid); luminal—ER/PR-positive, HER2-negative; luminal A—ER/PR-positive, Ki67 ≤ 15%, HER2-negative; luminal B—ER/PR-positive, Ki67 > 15%, HER2-negative; luminal HER2—ER/PR-positive, HER2-positive; HER2—ER/PR-negative, HER2 positive; TN NOS—ER/PR-negative, HER2-negative, CK5/6/CK17/CK14/EGFR-negative; basal—ER/PR-negative, HER2-negative, CK5/6/CK17/CK14 and/or EGFR-positive.

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
