# Peer review of "Low Prevalence of HER2-Positive Breast Carcinomas among Screening Detected Breast Cancers"

_cancers, 2020, doi:10.3390/cancers12061578_

Round 1
Reviewer 1 Report
Thank you for the opportunity to review a scientific article titled as “Low prevalence of HER2-positive breast carcinomas among screening detected breast cancers”. In this study, authors have found out the HER2-postive invasive breast cancer incidence among screening (SDBC) is lower than non-SDBC (NSDBC) from the two cohorts of regional screening programs in Spain. And also, it was reported that the prognosis of SDBC is better than NSDBC regarding to the relapse free survival and breast cancer specific survival. Then, to clear the discrepancy in incidence of HER2-positive breast cancer, they have collected and analyzed same previous studies in detail. As emphasized by the authors, it was a strength to use a control group in the same age range that those attending screening programs. It seems to be well analyzed and written to inform to readers, the important epidemiological knowledge and a convincing message to consider future strategies in breast cancer fields.
For more satisfied for the publication, I have some minor comments as below.
- In table 5, in the column of tumor characteristics, you should change “RE” to “ER” and “RP” to “PgR”.
- To discuss the differences between SDBC and NSDBC, the fact that the invasive tumor size is smaller in SDBC is considered to be an important issue. I think it is better to consider whether this fact influences the difference in distribution of tumor biology. I agree with the authors’ comments that the difference in the frequency of HER2-positive breast cancer between SDBC and NSDBC has a very important impact on patients. This is because, in addition to surgery, systemic therapies such as chemotherapy and anti-HER2 therapy are essential to conquer cancer and these standard therapies have some burden for patients. Penetration of screening program suggests that these breast cancer patients may be less frequent. I would like to recommend you should additionally analyses in specialized in HER2-positive breast cancer and insert the results in this manuscript. You may compare the prognosis of HER2-postive SDBC and HER2-positive NSDBC. Also, make a distribution chart of tumor size for each group. If the prognosis of both groups were same and the size distributions tend to be the same, this will lead to new opportunities to consider changes in biology in cancer progression. I guess there is basically a time lead bias in comparison SDBC and NSDBC, and may reflect the characteristics of early detection in the process of cancer development.
- It is better you should add some suggestions why the incidence of HER2-positive is lower among SDBC from the view of the biological aspects.
Author Response
We would like to thank Reviewer 1 for your helpful suggestions.
As it was suggested, English language and style have been checked by a certified English teacher.
- In table 5, in the column of tumor characteristics, you should change “RE” to “ER” and “RP” to “PgR”.
“RE” has been changed to “ER” and “RP” to “PR”.
- To discuss the differences between SDBC and NSDBC, the fact that the invasive tumor size is smaller in SDBC is considered to be an important issue. I think it is better to consider whether this fact influences the difference in distribution of tumor biology. I agree with the authors’ comments that the difference in the frequency of HER2-positive breast cancer between SDBC and NSDBC has a very important impact on patients. This is because, in addition to surgery, systemic therapies such as chemotherapy and anti-HER2 therapy are essential to conquer cancer and these standard therapies have some burden for patients. Penetration of screening program suggests that these breast cancer patients may be less frequent. I would like to recommend you should additionally analyses in specialized in HER2-positive breast cancer and insert the results in this manuscript. You may compare the prognosis of HER2-postive SDBC and HER2-positive NSDBC. Also, make a distribution chart of tumor size for each group. If the prognosis of both groups were same and the size distributions tend to be the same, this will lead to new opportunities to consider changes in biology in cancer progression. I guess there is basically a time lead bias in comparison SDBC and NSDBC, and may reflect the characteristics of early detection in the process of cancer development.
We have included data regarding tumor size, stage and prognosis of HER2-positive SDBC and HER2-positive NSDBC as supplementary information (Table S4: Cox model for histopathological features for disease-free survival and cancer-specific free survival within the HER2-positive breast cancer tumors). We have also included a paragraph in the Discussion section suggesting that differences in the incidence between both groups is probably related to a lead time bias (page 17, line 273).
- It is better you should add some suggestions why the incidence of HER2-positive is lower among SDBC from the view of the biological aspects.
From a biological point of view, we suggested that HER2-positive tumors are less frequently detected in screening programs due to their higher proliferation rate (page 18, line 328). Supplementary table S3 shows Ki67 proliferation index among different subrogate phenotypes.
Reviewer 2 Report
The manuscript by Angeles López-García and al. reports the low prevalence of HER2-positive breast cancers (BC) in screening-detected BC. The authors analyzed two independent hospital-based series and compared screen-detected and non-screen-detected BC diagnosed in 50-69 year old women. Results were compared and integrated with data obtained by a literature review.
- Data reported in Tables 1-6 and Figure 1 are all referred to PDP series and this should to be clearly indicated in the manuscript. The DEPRECAM series was used only for the determination of HER2 and HER2-related subsets.
- The controversial data on the prevalence of HER2 are often dependent on the technical differences in its determination. In the present work, the prevalence of HER2 and the consequent proportion of HER2-related subsets are quite different in the PDP and DEPRECAM series. Considering that the reagents used for the determination of ER, PR and HER2 were different in the 2 series (Table S2), the concordance of the IHC determination of ER, PR and especially HER2 in the 2 hospitals needs to be assessed to allow a reliable comparison.
- The search strategy, the selection criteria and the methodology and analysis of the systematic review are lacking and need to be detailed in Material and Methods.
Minor comments
The definition of Luminal B lacks of “HER2 negative” in Tables 1, 4, 9 and S1.
Author Response
We would like to thank Reviewer 2 for your helpful suggestions.
- Data reported in Tables 1-6 and Figure 1 are all referred to PDP series and this should to be clearly indicated in the manuscript. The DEPRECAM series was used only for the determination of HER2 and HER2-related subsets.
We have included that the series presented in Table 1-6 and Figure 1 are referred to PDP.
- The controversial data on the prevalence of HER2 are often dependent on the technical differences in its determination. In the present work, the prevalence of HER2 and the consequent proportion of HER2-related subsets are quite different in the PDP and DEPRECAM series. Considering that the reagents used for the determination of ER, PR and HER2 were different in the 2 series (Table S2), the concordance of the IHC determination of ER, PR and especially HER2 in the 2 hospitals needs to be assessed to allow a reliable comparison.
We have performed a concordance analysis on 58 tumors included on 2 TMAs sections. Data regarding concordance have been included as Supplementary table S2: Concordance analysis between HER2 status in both hospitals. A paragraph on this topic has also been included in the Discussion section (page 17, line 297).
- The search strategy, the selection criteria and the methodology and analysis of the systematic review are lacking and need to be detailed in Material and Methods.
We have included a paragraph in the Materials and Methods section briefly describing the search strategy of the articles included in this study (page 21, line 423).
- Minor comments: The definition of Luminal B lacks of “HER2 negative” in Tables 1, 4, 9 and S1.
The definition of Luminal B has been completed in such Tables.